# Heterogeneous Incremental Learning for Dense Prediction: Advancing Knowledge Retention via Self-Distillation

## Abstract

Incremental Learning (IL) aims to preserve knowledge acquired from previous tasks while incorporating knowledge from a sequence of new tasks. However, most prior work explores only streams of homogeneous tasks (*e.g.*, only classification tasks) and neglects the scenario of learning across heterogeneous tasks that possess different structures of outputs. In this work, we formalize this broader setting as heterogeneous incremental learning (HIL). Departing from conventional IL, the task sequence of HIL spans different task types, and the learner needs to retain heterogeneous knowledge for different output space structures. To instantiate the HIL, we focus on HIL in the context of dense prediction (HIL4DP), a more realistic and challenging scenario. To this end, we propose the Heterogeneity-aware Incremental Self-Distillation (HISD) method, an exemplar-free approach that preserves previously gained heterogeneous knowledge by self-distillation incrementally. HISD comprises two complementary components: a distribution-balanced loss to alleviate the global imbalance of prediction distribution and a salience-guided loss that concentrates learning on informative edge pixels extracted with the Sobel operator. Extensive experiments demonstrate that the proposed HISD significantly outperforms existing IL baselines in this new scenario.

## 1 Introduction

Incremental learning (IL), also known as continual learning, has garnered significant attention since it holds the potential to continually adapt to a sequence of new tasks from the data stream (Dohare et al., 2024; Lee et al., 2024; Zhuang et al., 2024). The primary objective of IL is to address the catastrophic forgetting problem (McCloskey & Cohen, 1989), which refers to the performance degradation on previously learned tasks after learning new tasks in the absence of historical data.

Previous IL methods (Zhao et al., 2024; Yang et al., 2024b) in the field of computer vision are primarily developed within the context of specific tasks (*e.g.*, classification-only or segmentation-only), limiting the applicability of IL methods to broader scenarios. Specifically, the IL setting often assumes the arrival of homogeneous tasks, overlooking real-world scenarios where heterogeneous tasks (*e.g.*, classification and regression tasks) emerge continuously. Furthermore, sequentially handling heterogeneous tasks, which requires the integration of heterogeneous knowledge, remains underexplored. Those limitations present challenges to traditional IL and necessitate extending IL to a novel scenario of heterogeneous incremental learning (HIL), in which the incoming tasks are heterogeneous (*e.g.*, a data stream with a mixture of regression and classification tasks).

To instantiate the HIL setting, in this paper, we focus on a fundamental class of computer vision problems (Yuan & Zhao, 2024), dense prediction (DP) tasks, under the setup of HIL. The primary goal of DP is to learn a mapping from input images to pixel-wise annotated labels (Kim et al., 2023), with heterogeneous label spaces for different DP tasks (*e.g.*, concrete class labels and continuous depth maps). Considering the high cost of annotation and data scarcity, previous works have jointly trained on DP tasks to achieve better performance (Vandenhende et al., 2021; Yang et al., 2024a; Wang et al., 2025). However, due to privacy concerns and temporal inconsistency in data collection (Zhou et al., 2023; Yang et al., 2025a; Xu et al., 2025), jointly training on multiple tasks becomes impractical. This motivates our exploration of the HIL for dense prediction (HIL4DP). In

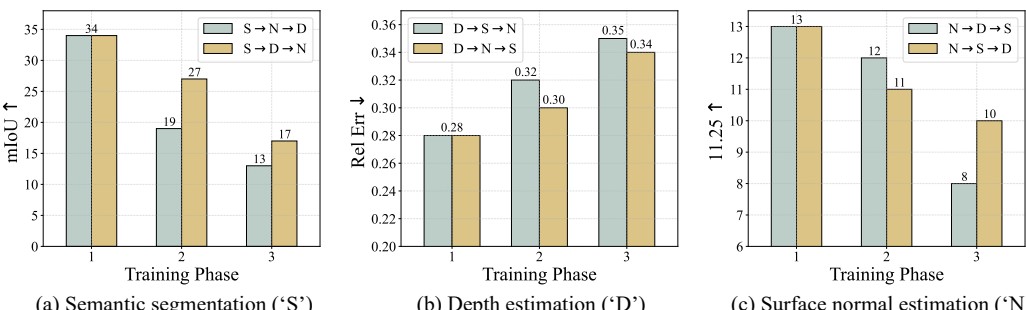

(a) Semantic segmentation ('S')  (b) Depth estimation ('D')  (c) Surface normal estimation ('N')

Figure 1: Vanilla training under HIL4DP. To assess the impact of catastrophic forgetting, we shuffle the learning sequences of three DP tasks. Each figure illustrates how the performance of a given task varies as the training phase proceeds, where the number in the horizontal axis denotes the task index in each sequence of three DP tasks. The performance metric is indicated above each column. The symbol ↑ (↓) signifies that a higher (lower) value denotes better performance.

this scenario, the input data across tasks originates from the same domain, yet the tasks encountered sequentially are heterogeneous DP tasks. Under this scenario, we investigate the presence of catastrophic forgetting by performing vanilla training on sequentially introduced task data, with the experimental settings described in Sec. 5.1. As shown in Fig. 1, all tasks suffer from catastrophic forgetting regardless of the learning sequences.

To mitigate the issue, we propose the Heterogeneity-aware Incremental Self-Distillation (HISD) method. HISD performs self-distillation in an exemplar-free manner, *i.e.*, without storing historical data. It maintains the heterogeneous knowledge learned from previous tasks by generating pseudo-labels to guide the knowledge retention. To improve the effectiveness of the pseudo-label guidance, we propose two novel loss functions in the HISD method. Firstly, a distribution-balanced incremental self-distillation (DB-ISD) loss is proposed to mitigate imbalanced pseudo-labels in dense prediction tasks (Jiao et al., 2018; Li et al., 2020; Ren et al., 2022; Zhong et al., 2023) by balancing the distribution of different semantic groups. Additionally, we use the geometric mean to smooth the self-distillation loss within each group, which reduces the noise in pseudo-labels. Second, a proposed salience-guided incremental self-distillation (SG-ISD) loss utilizes the Sobel operator (Sobel, 2014) to extract the semantic boundaries of predictions, thereby emphasizing the loss of pixels near semantic boundaries to maintain previous knowledge more effectively.

In summary, the contributions of this paper are three-fold. *a)* We introduce a new scenario, heterogeneous incremental learning (HIL), and emphasize its unique challenges related to heterogeneous tasks and knowledge, in contrast to traditional IL. In particular, we investigate a more realistic case: HIL for dense prediction (HIL4DP). *b)* We propose the HISD method, which consists of two components: DB-ISD and SG-ISD. *c)* Comprehensive experiments across diverse datasets in the HIL4DP scenario validate the effectiveness of the proposed HISD approach, indicating that HISD mitigates catastrophic forgetting in the HIL4DP scenario more effectively than traditional IL baselines.

## 2 RELATED WORK

**Incremental learning.** Incremental learning (IL), also known as continual learning, aims to enable models to continually acquire new knowledge from streaming data while mitigating catastrophic forgetting of previously learned knowledge. Traditional works on IL can be broadly classified into three categories (De Lange et al., 2021): replay methods, which store exemplars and replay historical data (Rebuffi et al., 2017; Aljundi et al., 2019; Buzzega et al., 2020), regularization methods, which introduce additional regularization terms (Kirkpatrick et al., 2017; Deng et al., 2021; Saha & Roy, 2023; Bhat et al., 2023), and parameter isolation methods (De Lange et al., 2021), which assign separate model parameters to each new task while masking parameters associated with previous tasks (Fernando et al., 2017; Konishi et al., 2023). While existing works extend IL to handle heterogeneity in terms of class attributes (Dong et al., 2023; Goswami et al., 2023), data distribution (Wuerkaixi et al., 2025), and model structures (Madaan et al., 2023), they primarily focus on single-task-type scenarios and ignore the heterogeneity of tasks. In contrast, we turn our attention to a more challenging scenario, where the learning process involves a series of tasks with heterogeneous outputs.

**Dense prediction.** Dense prediction (DP) tasks, such as semantic segmentation, depth estimation, and surface normal prediction, are fundamental in computer vision (Cordts et al., 2016; Vandenhende et al., 2021; Zuo et al., 2022). Those tasks involve per-pixel discrete label or continuous value prediction, requiring fine-grained feature extraction and globally consistent outputs. In general, they pose greater challenges than image-level prediction tasks (Zuo et al., 2022). To achieve better performance, various methods are designed for DP tasks (Ronneberger et al., 2015; Chen et al., 2018; Ranftl et al., 2021). Although those supervised methods achieve remarkable performance, they rely heavily on large-scale, high-quality pixel-level annotated data, which is costly to obtain (Yang et al., 2025b; Xu et al., 2025). To mitigate data scarcity, multi-task learning methods (Zuo et al., 2022; Ye & Xu, 2024; Wang et al., 2025) are proposed to simultaneously learning DP tasks within a single model. However, practical constraints (*e.g.*, data privacy, limited resources, and sequential data collection (Zhou et al., 2023; Zhao et al., 2024)) render joint training across tasks difficult, which motivates the study of DP tasks under the proposed new scenario HIL.

**IL for dense prediction.** Existing IL scenarios for DP tasks are typically tailored for the specific tasks, including incremental depth estimation (IDS) and continual semantic segmentation (CSS). The former focuses on enabling continuous depth estimation in emerging domains (Yang et al., 2024b), while the latter concerns with segmentation in incremental shift along class and domain directions (Toldo et al., 2024). We differ from these scenarios in two primary aspects. First, compared with previous task-specific scenarios, HIL4DP not only reduces the risk of overfitting on the specific task (Zhang & Yang, 2021), but holds the potential of learning knowledge from related vision tasks (Vandenhende et al., 2021). Second, while effective within their scope, these task-specific methods rely on homogeneous, task-specific information, making them unsuitable for a sequence of heterogeneous tasks (*i.e.*, HIL4DP). For instance, discrete class labels (Gong et al., 2024; Yin et al., 2025) or classification probabilities (Douillard et al., 2021; Toldo et al., 2024) commonly used in CSS are not available in regression-based tasks. Similarly, domain-aware solutions in IDS (Hu et al., 2023; Yang et al., 2024b) are designed to address challenges such as domain shift and depth spatial variations. Consequently, they are fundamentally inapplicable to the HIL4DP setting.

## 3 PROBLEM DEFINITION

Conventional IL assumes that all tasks share the same output structure. However, real-world applications such as dense prediction (DP) demand a learner capable of handling heterogeneous tasks whose outputs include class labels, depth maps, surface normal vectors, and more. We formalize this more general and challenging scenario as HIL in Sec. 3.1, and then formalize the more realistic scenario HIL4DP in Sec. 3.2. Finally, we analyze the challenges of HIL4DP in Sec. 3.3.

### 3.1 HETEROGENEOUS INCREMENTAL LEARNING (HIL)

In the HIL setting, let $\mathcal{T} = \{\mathcal{T}_t\}_{t=1}^T$ be a sequence of $T$ heterogeneous tasks , where all tasks share a common input space $\mathcal{X}$ but each task $\mathcal{T}_t$ has its own output space $\mathcal{Y}_t$ as

$$\mathcal{T}_t = \{(\mathcal{X}, \mathcal{Y}_t)\}, \qquad t = 1, \ldots, T. \tag{1}$$

Due to the heterogeneity between tasks, the output space of each task varies, *e.g.*, continuous or discrete outputs. Each task $\mathcal{T}_t$ has its corresponding training dataset $\mathcal{D}_t = \{(x, y) | x \in \mathcal{X}, y \in \mathcal{Y}_t\}$, where $(x, y)$ refers to the input and its corresponding label, respectively. Here, we assume that training instances in different tasks have no overlap. Note that during the $t$-th training phase, only the corresponding dataset $\mathcal{D}_t$ of task $\mathcal{T}_t$ is available, and the datasets of other tasks are unavailable.

The objective of HIL is to design a unified heterogeneous incremental learner, $\mathcal{F} : \mathcal{X} \to \bigcup_{i=1}^T \mathcal{Y}_i$, capable of incrementally adapting to the sequence of tasks $\mathcal{T}$. Specifically, at the $t$-th training phase, the learner is expected to accurately predict outputs over the cumulative heterogeneous tasks $\mathcal{T}_{1:t}$. This requires the learner to retain knowledge acquired during the previous training phases without access to prior data.

### 3.2 HIL FOR DENSE PREDICTION (HIL4DP)

The above problem setup is universal and holds the potential to benefit a wide range of downstream heterogeneous tasks. In this paper, we focus on a challenging and realistic scenario involving a

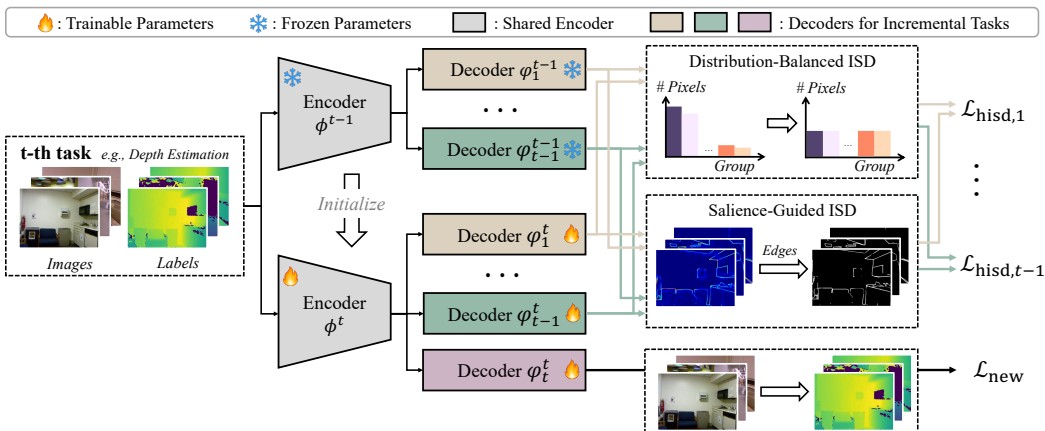

Figure 2: The training pipeline of the proposed HISD method in the $t$-th training phase. The HISD method uses the distribution-balanced ISD and salience-guided ISD to mitigate forgetting of previous tasks $\mathcal{T}_j$ $(j < t)$, all of which are calculated on the pseudo-labels generated by the frozen teacher model $\mathcal{F}_j^{t-1}$. Adapting to the new task $\mathcal{T}_t$ is achieved by the task-specific loss function $\mathcal{L}_{\text{new}}$.

sequence of heterogeneous dense prediction tasks. Each task $\mathcal{T}_t$ corresponds to a distinct dense prediction task (*e.g.*, semantic segmentation, depth estimation, or surface normal prediction) with a unique output space (*e.g.*, class labels, depth maps, or surface normal vectors).

Formally, the input and output spaces of task $\mathcal{T}_t$ are defined as

$$\mathcal{T}_t = \{(\mathcal{X}, \mathcal{Y}_t)|\mathcal{X} \subseteq \mathbb{R}^{C^{\text{in}} \times H \times W}, \mathcal{Y}_t \subseteq \mathbb{R}^{C_t^{\text{out}} \times H \times W}\}, \tag{2}$$

where $H, W$ denotes the spatial dimensions (*i.e.*, the height and width of input images), $C^{\text{in}}$ denotes the number of input channels (*e.g.*, 3 for RGB images), $C_t^{\text{out}}$ denotes the number of output channels for the $t$-th task $\mathcal{T}_t$, which varies across tasks (*e.g.*, 1 for the depth estimation task or 3 for the surface normal estimation task). Moreover, $\mathcal{Y}_t$ is continuous for some tasks (*e.g.*, depth estimation and surface normal estimation) or discrete for some tasks (*e.g.*, semantic segmentation), while all tasks share the same input domain. A detailed comparison between the proposed HIL4DP scenario and existing IL subcategories is provided in Appendix A.

### 3.3 CHALLENGES

HIL4DP poses challenges that extend beyond conventional IL. It involves sequentially learning *heterogeneous tasks* with distinct objectives and outputs, resulting in a more complex and challenging process. These tasks rely on *heterogeneous knowledge* (*e.g.*, 3D scene understanding in depth estimation versus semantic structure in segmentation), making it difficult to balance knowledge retention and forgetting. Furthermore, the pixel-level nature of dense prediction requires preserving *fine-grained information* while maintaining globally consistent outputs. Additional discussion of these challenges posed by HIL4DP is provided in Appendix B.

According to the above analysis, the challenges in the HIL4DP scenario can be attributed to the unique nature of task heterogeneity and further compounded by the added complexity of DP tasks. In the next section, we propose a method to handle those challenges.

## 4 METHODOLOGY

In this section, we introduce the proposed HISD method for the HIL4DP setting.

### 4.1 OVERVIEW

**Architecture.** As illustrated in Fig. 2, we employ a task-shared encoder to acquire knowledge from a sequence of tasks and capture fine-grained features from images. Given the heterogeneity across tasks, a task-specific decoder is used per task. Formally, the learner $\mathcal{F}$ during the $t$-th training phase

comprises: 1) a task-shared encoder $f_{\phi^t} : \mathcal{X} \to \mathbb{R}^d$ parameterized by $\phi^t$ that generalizes across tasks; 2) a set of task-specific decoders parameterized by $\{\varphi_i^t\}_{i=1}^t$, where each $g_{\varphi_i^t} : \mathbb{R}^d \to \mathcal{Y}_i$ maps hidden features to the specific output space of task $\mathcal{T}_i$. To simplify notation, the prediction function of task $\mathcal{T}_j$ is defined as $\mathcal{F}_j^t(\cdot) = \mathcal{F}(\cdot; \phi^t, \varphi_j^t) : \mathcal{X} \to \mathcal{Y}_j$, and the parameters of learner $\mathcal{F}$ at the training phase $t$ is denoted as $\Phi^t = \{\phi^t, \{\varphi_j^t\}_{j=1}^t\}$.

**Incremental Self-Distillation (ISD).** To enable continuous knowledge integration from new DP tasks, the parameters $\Phi^t$ are initialized from $\Phi^{t-1}$ with the expanded task-specific decoder $\varphi_t^t$ to adapt to task $\mathcal{T}_t$. Then, $\Phi^t$ is trained on the new DP task $\mathcal{T}_t$ using the task-specific training loss $\mathcal{L}_{\text{new}}$ (*e.g.*, $L_1$ loss for depth estimation task and cross-entropy loss for semantic segmentation tasks) via supervised learning, while retaining the previous knowledge via self-distillation loss $\mathcal{L}_{\text{dis}}$.

However, as shown in Fig. 1, vanilla training on a new task $\mathcal{T}_t$ leads to catastrophic forgetting of previous tasks in the HIL4DP scenario. To mitigate this problem, a simple method is to employ the self-distillation (Pham et al., 2022). Specifically, during the $t$-th training phase, the previous learners $\{\mathcal{F}_i^{t-1}\}_{i=1}^{t-1}$ trained on previous tasks $\mathcal{T}_{1:t-1}$ are treated as the teacher model, while the current learners $\{\mathcal{F}_i^t\}_{i=1}^{t-1}$ being trained on the new task $\mathcal{T}_t$ serve as the student model. To retain prior knowledge of each task $\mathcal{T}_j$, we introduce the distillation loss function $\mathcal{L}_{\text{dis},j}$ to align the prediction of the student model on $\mathcal{D}_t$ with the pseudo-label generated by the teacher model. Since datasets $\mathcal{D}_{1:t-1}$ of previous tasks are inaccessible and every task shares a common input space, the pseudo-labels for previous tasks can be generated on the dataset $\mathcal{D}_t$ of the new task. Formally, the total training loss to train $\Phi^t$ is formulated as

$$\mathcal{L} = \alpha \underbrace{\sum_{(x,y)\in\mathcal{D}_t} \frac{1}{(t-1)|\mathcal{D}_t|} \sum_j^{t-1} \bar{\mathcal{L}}_{\text{dis},j}(\mathcal{F}_j^t(x), \mathcal{F}_j^{t-1}(x))}_{\mathcal{L}_{\text{dis}}} + \underbrace{\frac{1}{|\mathcal{D}_t|} \sum_{(x,y)\in\mathcal{D}_t} \bar{\mathcal{L}}_t(\mathcal{F}_t^t(x), y)}_{\mathcal{L}_{\text{new}}}, \quad (3)$$

where $\bar{\mathcal{L}}_{\text{dis},j}$ is the task-specific distillation loss function of task $\mathcal{T}_j$, $\bar{\mathcal{L}}_t$ is the task-specific loss function of task $\mathcal{T}_t$, $|\mathcal{D}_t|$ denotes the number of samples in the dataset $\mathcal{D}_t$, and $\alpha$ is the hyperparameter to control the impact of the distillation loss $\mathcal{L}_{\text{dis}}$.

However, as illustrated in the next sections, naive distillation in Eq. (3) yields limited gains due to imbalanced pseudo-label distributions and insufficient focus on salient regions. To address those issues, we further propose the HISD method, consisting of two loss components for each task $j$: distribution-balanced incremental self-distillation (DB-ISD) loss $\mathcal{L}_{\text{db},j}$ and salience-guided incremental self-distillation (SG-ISD) loss $\mathcal{L}_{\text{sg},j}$. Thus, the total training loss to learn $\Phi^t$ in the proposed method is formulated as

$$\mathcal{L} = \frac{\alpha}{2(t-1)} \sum_{j=1}^{t-1} \underbrace{\sum_{(x,y)\in\mathcal{D}_t} \frac{1}{|\mathcal{D}_t|} (\mathcal{L}_{\text{db},j}(x) + \mathcal{L}_{\text{sg},j}(x))}_{\mathcal{L}_{\text{hisd},j}} + \mathcal{L}_{\text{new}}. \quad (4)$$

The details of DB-ISD and SG-ISD are introduced in the following sections.

## 4.2 DISTRIBUTION-BALANCED INCREMENTAL SELF-DISTILLATION (DB-ISD)

To preserve previous knowledge while learning new tasks, the teacher model generates pseudo-labels on the new training data to revise the heterogeneous knowledge of previous tasks. However, we observe that the distribution of generated pseudo-labels is imbalanced. To address the imbalance issue, we propose the DB-ISD method, which first partitions image pixels into semantic groups and then balances their respective contributions.

**Imbalance issue.** Generally, DP tasks can be categorized into pixel-level classification and pixel-level regression tasks. To illustrate the imbalance phenomenon across these two types of DP tasks, Fig. 3 visualizes the distribution of pseudo-labels generated by the learner after the first training phase on raw images from the new task data, exhibiting an imbalance pixel-wise distribution of class labels for the classification task (*e.g.*, semantic segmentation in Fig. 3(b)) and values for the regression task (*e.g.*, depth estimation in Fig. 3(c)). This phenomenon is widespread across different tasks rather than being limited to our experiments (Ge et al., 2024), posing a risk to effective knowledge retention during the learning phase of the new task (Jiao et al., 2018).

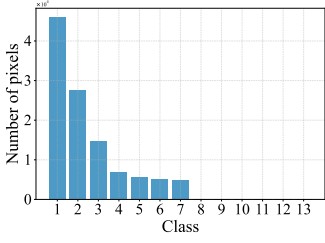 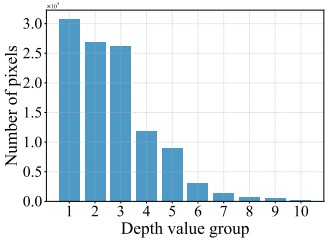

(a) An example of raw images     (b) Semantic segmentation task     (c) Depth estimation task

Figure 3: An illustration of the distribution imbalance in pseudo-labels. The number of pixels in the semantic segmentation task is counted per class. In the depth estimation task, we divide the range of pseudo-labels given by the teacher model into ten equal intervals, each of which is a group, and then the ten groups are sorted based on the number of pixels in each group.

**Group partition.** A group is defined as a collection of pixels that share similar semantics, as decided by their pseudo-labels. For each task $\mathcal{T}_j$, the generated pixel-level pseudo-labels $\mathcal{F}_j^{t-1}(x) \in \mathbb{R}^{H \times W}$ on the image $x \in \mathcal{D}_t$ are divided into $C_j$ non-overlapping groups. For a pixel-level classification task $\mathcal{T}_j$ in DP, a group corresponds to a class. The number of groups $C_j$ equals the number of classes. Concretely, for each class $c \in \{i\}_{i=1}^{C_j}$, we construct binary masks $M_{c,j}^x \in \{0,1\}^{H \times W}$ that indicates the presence of class $c$: $M_{c,j}^x[m,n] = \mathbb{I}(\mathcal{F}_j^{t-1}(x)[m,n] = c)$, where $\mathbb{I}(\cdot)$ is the indicator function, $m \in \{1,\ldots,H\}$, $n \in \{1,\ldots,W\}$ denote the indices of the mask, and $A[m,n]$ for a matrix $A$ denotes the $(m,n)$-th entry in $A$. For a pixel-level regression task $\mathcal{T}_j$ in DP, we first obtain a scalar value per pixel by averaging across the channel dimension of size $C_j^{\text{out}}$. The resulting continuous values are then min–max normalized (Bishop & Nasrabadi, 2006) into the interval [0,1] and binarized into two groups ($C_j = 2$), *i.e.*, foreground and background (Ge et al., 2024), using a threshold $\tau \in (0,1)$. This yields two masks: $M_{1,j}^x[m,n] = \mathbb{I}(\mathcal{F}_j^{t-1}(x)[m,n] < \tau)$, and $M_{2,j}^x[m,n] = \mathbb{I}(\tau \le \mathcal{F}_j^{t-1}(x)[m,n])$.

**Loss function.** Inside each group, we compute the *geometric mean* of the per-pixel self-distillation loss to mitigate the impact of inaccuracies and noise (Tao et al., 2008). The group losses are then averaged arithmetically, ensuring that each group contributes equally to the training objective. Given an input image $x$, the loss function $\mathcal{L}_{\text{db},j}$ of DB-ISD for each task $\mathcal{T}_j$ during the $t$-th training phase can be expressed as

$$\mathcal{L}_{\text{db},j}(x) = \sum_{c=1}^{C_j} \frac{1}{C_j} \left( \prod_{(m,n) \in I_{c,j}(x)} \mathcal{L}_{\text{dis},j}\big(\mathcal{F}_j^t(x), \mathcal{F}_j^{t-1}(x)\big)[m,n] \right)^{\frac{1}{|I_{c,j}(x)|}}, \qquad (5)$$

where $\mathcal{L}_{\text{dis},j}$ denotes the per-pixel self-distillation loss of task $\mathcal{T}_j$, $I_{c,j}(x) = \{(m,n)|M_{c,j}^x[m,n] > 0\}$ is a set of indices that the corresponding pixel belongs to the group $c$ for the image $x$ of task $\mathcal{T}_j$, and $|I_{c,j}(x)|$ denotes the number of elements in $I_{c,j}(x)$.

### 4.3 SALIENCE-GUIDED INCREMENTAL SELF-DISTILLATION (SG-ISD)

In dense prediction tasks, a substantial amount of informative signal resides around semantic boundaries or sharp value transitions (Zhu et al., 2020; Zuo et al., 2022). Preserving this information in HIL4DP effectively enhances the retention of heterogeneous knowledge. Hence, we introduce a complementary salience-guided loss to ensure that the model retains information in these pixels.

**Salient-pixel extraction.** To enhance distillation, the SG-ISD loss focuses on the edges of the pixel-wise loss map, corresponding to pixels with sharp variations that carry the most informative signals. Though ground-truth edges are unavailable, an edge set can be obtained by identifying pixels where the value changes significantly between adjacent pixels (Vincent et al., 2009). We first calculate a pixel-wise loss map between the frozen teacher model $\mathcal{F}_j^{t-1}$ and the student model $\mathcal{F}_j^t$ as

$$\mathbf{I}_j(x) = \mathcal{L}_{\text{dis},j}(\mathcal{F}_j^t(x), \mathcal{F}_j^{t-1}(x)) \in \mathbb{R}^{H \times W}, \qquad (6)$$

where $\mathbf{I}_j(x)$ denotes the pixel-wise self-distillation loss map of input $x$ for task $\mathcal{T}_j$. To localize sharp spatial transitions in $\mathbf{I}_j(x)$, we apply the Sobel operator (Sobel, 2014), a discrete differenti-

ation operator that approximates the gradients of the image intensity function. With the horizontal convolution kernel defined as $\mathbf{G}_h = [1, 2, 1]^\top [1, 0, -1]$, and the vertical convolution kernel defined as $\mathbf{G}_v = [1, 0, -1]^\top [1, 2, 1]$, the Sobel operator conduct the gradient approximation as

$$\mathbf{G}_j(x) = \sqrt{(\mathbf{G}_h * \mathbf{I}_j(x))^2 + (\mathbf{G}_v * \mathbf{I}_j(x))^2}, \tag{7}$$

where $*$ denotes the convolution operator and the superscript $(\cdot)^2$ denotes the elementwise square operation. The edge set $\mathbf{P}_j$ is then selected by thresholding the gradient magnitude map $\mathbf{G}_j$ as

$$\mathbf{P}_j(x) = \{(m, n) \mid \mathbf{G}_j(x)[m, n] > k\}, \tag{8}$$

where $k$ is a hyperparameter controlling the necessary gradient intensity to constitute an edge.

**Loss function.** We accumulate the pixel-wise loss over the extracted salient edge set only:

$$\mathcal{L}_{\mathrm{sg},j}(x) = \sum_{(m,n) \in \mathbf{P}_j(x)} \frac{1}{|\mathbf{P}_j(x)|} \mathbf{I}_j(x)[m, n], \tag{9}$$

where $|\mathbf{P}_j(x)|$ denotes the number of elements in $\mathbf{P}_j(x)$.

To summarize, by plugging Eqs. (5) and (9) into Eq. (4), we obtain the objective function of the proposed HISD method in the $t$-th training phase. By balancing group contributions and emphasizing salient boundaries, the proposed HISD method provides an effective defense against forgetting.

## 5 EXPERIMENTS

### 5.1 EXPERIMENTAL SETUP

**Benchmarks.** We empirically evaluate the performance of the proposed method in the HIL4DP scenario under four well-established and practical DP benchmarks (Zhang et al., 2025; Wang et al., 2025), including *CityScapes* (Cordts et al., 2016), *NYUv2* (Silberman et al., 2012), *PASCAL-Context* (Everingham et al., 2010), and *Taskonomy* (Zamir et al., 2018), for scenarios involving 2, 3, 4, and 10 heterogeneous tasks. The task sequences are randomly selected, and the training data is evenly divided across tasks without overlap. Evaluation is performed using the full test set. During the training phase of each task, the labels of other tasks are inaccessible. Additional benchmark details can be found in Appendix C.1.

**Evaluation metrics.** Due to the heterogeneity of task outputs, different evaluation metrics are required to assess model performance across tasks. However, the disparity among these metrics makes it difficult to compare overall performance using simple averaging. Thus, following the setup in (Maninis et al., 2019), we adopt the average of the relative improvement over the vanilla training across tasks after the $t$-th training phase as the overall evaluation metric, defined as

$$\Delta_b^t = \frac{1}{t} \sum_{i=1}^{t} \frac{1}{M_i} \sum_{j=1}^{M_i} \frac{(-1)^{s_{i,j}} (E_{i,j}^m - E_{i,j}^b)}{E_{i,j}^b}. \tag{10}$$

Here, $t$ is equal to the number of learned tasks, $M_i$ denotes the number of metrics for task $\mathcal{T}_i$. $E_{i,j}^m$ and $E_{i,j}^b$ denote the performance of the method $m$ and the vanilla training for the $j$-th metric in task $\mathcal{T}_i$, respectively. $s_{i,j}$ is set to 1 if a lower value indicates better performance in terms of the $j$-th metric in task $\mathcal{T}_i$ and otherwise 0. $\Delta_b^T$ denotes the final performance after the last training phase $T$, while $\bar{\Delta}_b^T = \frac{1}{T} \sum_{j=1}^{T} \Delta_b^j$ denotes the average performance across $T$ training phases.

**Comparison methods.** In the HIL4DP setup, we compare the proposed method HISD with IL methods applicable to this scenario. Specifically, baseline methods include EWC (Kirkpatrick et al., 2017), LWF (Li & Hoiem, 2017), iCaRL (Rebuffi et al., 2017), DER (Buzzega et al., 2020), SPG (Konishi et al., 2023), and SGP (Saha & Roy, 2023). For the replay-based baseline methods, DER and iCaRL, we store and replay the pixel-wise predictions to ensure fair comparison with HISD. In addition to these IL methods, we also establish two extreme baselines for comparison, including *a)* Vanilla training, which involves sequentially training tasks, *b)* Joint training, where all tasks are trained simultaneously using the complete dataset, which serves as the upper bound. For

Table 1: Performance on 3 tasks (*i.e.*, 13-class semantic segmentation, depth estimation, and surface normal prediction) after the last training phase of the *NYUv2* dataset across different encoders. The best results for each task are shown in **bold**. ↑(↓) means that the higher (lower) the value, the better the performance.

| | Method | Segmentation | | Depth | | Surface Normal | | | | | $\Delta_b^T \uparrow$ |
| | | | | | | Angle Distance | | Within $t°$ | | | |
| | | mIoU↑ | Pix Acc↑ | Abs Err↓ | Rel Err↓ | Mean↓ | Median↓ | 11.25↑ | 22.5↑ | 30↑ | |
|---|---|---|---|---|---|---|---|---|---|---|---|
| *ResNet-18* | Vanilla training | 17.49 | 46.81 | 0.9609 | 0.3328 | 32.45 | 26.92 | 20.72 | 42.56 | 54.73 | +0.00% |
| | Joint training | 41.84 | 66.14 | 0.5793 | 0.2201 | 31.53 | 25.78 | 22.38 | 44.54 | 56.36 | +40.83% |
| | EWC | 32.17 | 57.21 | 0.9586 | 0.3493 | 37.52 | 33.08 | 13.47 | 33.37 | 45.35 | +24.39% |
| | iCaRL | 21.78 | 53.00 | 1.3093 | 0.4561 | 33.07 | 27.73 | 19.45 | 41.20 | 53.43 | −4.82% |
| | LwF | 31.51 | 57.37 | 0.8986 | 0.3345 | 37.06 | 32.09 | 13.89 | 34.66 | 46.81 | +24.74% |
| | DER | 21.90 | 53.10 | 1.2735 | 0.4422 | 33.09 | 27.74 | 19.36 | 41.18 | 53.43 | −3.31% |
| | SPG | 18.10 | 48.15 | 0.8801 | 0.3019 | **32.57** | **26.92** | **20.80** | **42.58** | **54.66** | +4.01% |
| | SGP | 21.34 | 49.75 | 0.9270 | 0.3181 | 32.87 | 27.15 | 19.99 | 42.15 | 54.31 | +6.53% |
| | HISD | **35.12** | **59.63** | **0.7410** | **0.2641** | 35.32 | 30.55 | 17.23 | 37.26 | 49.12 | **+32.74%** |
| *ResNet-50* | Vanilla training | 18.26 | 50.85 | 0.8305 | 0.2725 | 28.21 | 21.93 | 26.74 | 50.99 | 63.08 | +0.00% |
| | Joint training | 47.78 | 71.03 | 0.4933 | 0.2149 | 28.10 | 22.24 | 25.32 | 50.43 | 62.95 | +44.36% |
| | EWC | 36.55 | 61.75 | 0.7321 | 0.2629 | 33.61 | 28.99 | 18.37 | 39.34 | 51.51 | +31.08% |
| | iCaRL | 28.08 | 57.41 | 0.9877 | 0.3549 | 30.52 | 25.47 | 22.08 | 44.75 | 57.36 | +7.12% |
| | LwF | 38.06 | **63.77** | 0.6505 | 0.2466 | 31.84 | 26.28 | 20.67 | 43.33 | 56.07 | +32.94% |
| | DER | 27.12 | 58.04 | 0.7383 | 0.2650 | 31.25 | 26.30 | 21.50 | 43.58 | 56.91 | +17.83% |
| | SPG | 19.77 | 51.40 | 0.7595 | 0.2626 | 28.38 | **22.29** | 25.79 | **50.41** | **62.87** | +4.07% |
| | SGP | 18.99 | 51.52 | 0.8368 | 0.2764 | **28.27** | 22.52 | 25.79 | 49.91 | 62.50 | +1.15% |
| | HISD | **38.70** | 63.70 | **0.6294** | **0.2369** | 32.55 | 27.38 | 19.66 | 41.66 | 53.96 | **+35.71%** |

Table 2: Performance on 10 tasks: semantic segmentation (Seg.), depth estimation (Dep.), surface normal estimation (Normal), edge-2D detection (E.-2D), reshading (Res.), keypoint-2D detection (K.-2D), edge-3D detection (E.-3D), Euclidean distance (E. D.), curvatures (Curv.), and keypoint-3D detection. (K.-3D ) on the *Taskonomy* dataset. The lower the loss value, the better the performance.

| Method | Seg. | Dep. | Normal | E.-2D | Res. | K.-2D | E.-3D | E. D. | Curv. | K.-3D | $\Delta_b^T \uparrow$ |
|---|---|---|---|---|---|---|---|---|---|---|---|
| Vanilla | 0.7282 | 0.2673 | 0.2869 | 0.1627 | 0.4658 | 0.5143 | 0.5214 | 0.1968 | 1.8655 | 0.3863 | +0.00% |
| Joint | 0.1615 | 0.1071 | 0.1281 | 0.1434 | 0.1487 | 0.2969 | 0.3244 | 0.1054 | 1.3501 | 0.3149 | +44.57% |
| EWC | **0.4701** | 0.2358 | 0.1945 | 0.1692 | **0.3109** | 0.5327 | 0.5525 | 0.2286 | 1.9835 | 0.4347 | +6.41% |
| iCaRL | 0.5704 | 0.2247 | 0.2214 | 0.1846 | 0.3386 | 0.5022 | 0.5126 | 0.1918 | **1.7441** | 0.3965 | +8.47% |
| LwF | 0.6482 | 0.2628 | 0.2484 | 0.1774 | 0.4718 | 0.5298 | 0.5137 | **0.1879** | 1.9212 | 0.4208 | +0.68% |
| DER | 0.5993 | 0.2872 | 0.2315 | **0.1518** | 0.4584 | 0.5068 | **0.4564** | 0.2674 | 1.8488 | 0.3762 | +1.94% |
| SPG | 0.6768 | 0.2775 | 0.2617 | 0.1707 | 0.4641 | 0.5481 | 0.5418 | 0.2008 | 1.9087 | **0.3537** | +0.11% |
| SGP | 0.6751 | 0.2856 | 0.2595 | 0.1668 | 0.4608 | 0.5266 | 0.5319 | 0.2009 | 1.8036 | 0.3765 | +0.79% |
| HISD | 0.5357 | **0.2110** | **0.1868** | 0.1541 | 0.3379 | **0.4955** | 0.5114 | 0.2187 | 1.7530 | 0.4120 | **+10.90%** |

fair comparison, grid searches for task-specific hyperparameters are performed. Introduction and hyperparameter details of baselines are shown in the Appendix C.2.

**Implementation details.** We adopt the DeepLabV3+ architecture (Chen et al., 2018). Specifically, a pre-trained *ResNet-18* (He et al., 2016) with dilated convolutions (Yu et al., 2017) is used as the task-shared encoder across all tasks, and task-specific decoders are Atrous Spatial Pyramid Pooling (Chen et al., 2018). To further assess the effectiveness of the proposed method, we also conduct experiments using the *ResNet-50* (He et al., 2016) as the encoder. Details of hyperparameters are provided in Appendix C.3.

## 5.2 RESULTS

Tab. 1 presents the results of the proposed HISD method on the *NYUv2* dataset, evaluated across various architectures. As can be seen, compared to baseline methods, the proposed HISD method yields superior average performance across tasks. Moreover, the HISD method consistently enhances results across different architectures, highlighting its robustness and generalizability.

We also present the results of the proposed HISD method in the 10-task scenario on the *Taskonomy* dataset. As shown in Tab. 2, the HISD method outperforms the baseline methods, achieving the lowest test loss across the largest number of individual tasks, as well as the best average performance across all tasks. These results further demonstrate the effectiveness of the proposed HISD method.

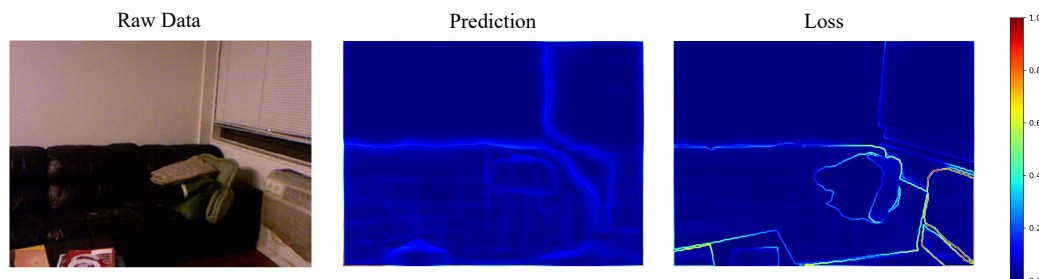

Figure 4: Visualization of the raw data (left), the gradient magnitude map of its prediction (middle), and the gradient magnitude map of the loss map (right).

## 5.3 Ablation study

We conduct ablations to assess the contribution of every component in HISD.

**Effectiveness of ISD.** We compared the incremental self-distillation (ISD) with several alternative training paradigms used for maintaining previous knowledge in the $t$-th$(t > 1)$ training phase: 1) While $\{\phi^t, \varphi_t^t\}$ is being trained on $\mathcal{L}_{\text{new}}$, the decoders $\{\varphi_t^j\}_{j=1}^{t-1}$ for the old tasks are updated using pseudo-labels. 2) After $\{\phi^t, \varphi_t^t\}$ is being trained on $\mathcal{L}_{\text{new}}$,

Table 3: Various training paradigms.

| Method | $\Delta_b^T \uparrow$ | $\bar{\Delta}_b^T \uparrow$ |
|---|---|---|
| Paradigm$_1$ | $-7.76\%$ | $-4.56\%$ |
| Paradigm$_2$ | $-6.65\%$ | $-5.23\%$ |
| Paradigm$_3$ | $-10.94\%$ | $-11.48\%$ |
| ISD | $+0.00\%$ | $+0.00\%$ |

the decoders $\{\varphi_t^j\}_{j=1}^{t-1}$ for the old tasks are updated using pseudo-labels. 3) Only updating the new decoder $\varphi_t^t$ with $\mathcal{L}_{\text{new}}$. All the training paradigms update the encoder and decoder of the first task on $\mathcal{L}_{\text{new}}$. As shown in Tab. 3, ISD consistently outperforms these alternatives.

**Effectiveness of DB-ISD.** We assess the effectiveness of the DB-ISD from three perspectives. First, as depicted in Tab. 4, removing DB-ISD from the baseline degrades the final metric by $2.84\%$ and the average metric by $2.35\%$. Second, substituting the geometric mean in Eq. (5) with the arithmetic mean ("HISD $w/$ Arithmetic" in Tab. 4) degrades performance, highlighting the advancement of the geometric mean. Third, for regression tasks, we compare our group partitioning with baselines that divide the value range into

Table 4: Ablation study.

| Method | $\Delta_b^T \uparrow$ | $\bar{\Delta}_b^T \uparrow$ |
|---|---|---|
| HISD | $+0.00\%$ | $+0.00\%$ |
| HISD $w/o\ \mathcal{L}_{\text{sg}}$ | $-1.27\%$ | $-0.43\%$ |
| HISD $w/o\ \mathcal{L}_{\text{db}}$ | $-2.84\%$ | $-2.35\%$ |
| HISD $w/$ Arithmetic | $-6.29\%$ | $-3.06\%$ |
| HISD ($\hat{C}_j$=5) | $-3.64\%$ | $-1.51\%$ |
| HISD ($\hat{C}_j$=10) | $-5.43\%$ | $-2.50\%$ |
| HISD ($\hat{C}_j$=15) | $-9.72\%$ | $-3.56\%$ |

equally sized intervals for group numbers $\hat{C}_j \in \{5, 10, 15\}$, where each interval has the same width of $\frac{1}{\hat{C}_j}$. As shown in Tab. 4, the proposed HISD outperforms all variants consistently.

**Effectiveness of SG-ISD.** As shown in Tab. 4, removing the salience-guided loss reduces the final and average metrics by $1.27\%$ and $0.43\%$, respectively, underscoring the importance of edge-aware focus to HISD. To assess the effectiveness of the Sobel operator applied to the loss map in Eqs. (6) and (7), we compare its gradient magnitude map with the one derived from the prediction map. As shown in Fig. 4, the gradient magnitude map of the loss map (right) produces more distinct edges than those from the prediction map (middle), making it better suited for identifying edge sets.

*Due to page limit, additional experiments, including results for different task sequences, additional model architectures, on other datasets, and an advantages analysis of the HIL4DP scenario by comparing it with training separate task-specific models are put in Appendix D and E.*

## 6 Conclusion

In this paper, we propose a novel incremental learning scenario named heterogeneous incremental learning (HIL), which brings unique challenges for traditional incremental learning. Specifically, we focus on the practical and challenging dense prediction tasks within the HIL scenario (HIL4DP). To address these unique challenges, we propose the heterogeneity-aware incremental self-distillation (HISD), which is composed of a prediction distribution balance and a salience-guided incremental self-distillation loss function. The comprehensive experimental results and ablation studies demonstrate the effectiveness of the proposed HISD method.

ETHICS STATEMENT

This work complies with the ICLR Code of Ethics. It involves no human subjects or animal experiments, relying solely on publicly available and authorized datasets. All authors confirm adherence to ethical guidelines and declare no conflicts of interest.

REPRODUCIBILITY STATEMENT

To ensure reproducibility, we provide the HISD code in the supplementary material. Full experimental details, including datasets, baselines, and hyperparameters, are presented in Appendix C, while Sec. 4 outlines the core algorithm. These resources support the reproduction of our main results.

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

## A COMPARISON WITH EXISTING INCREMENTAL LEARNING SCENARIOS

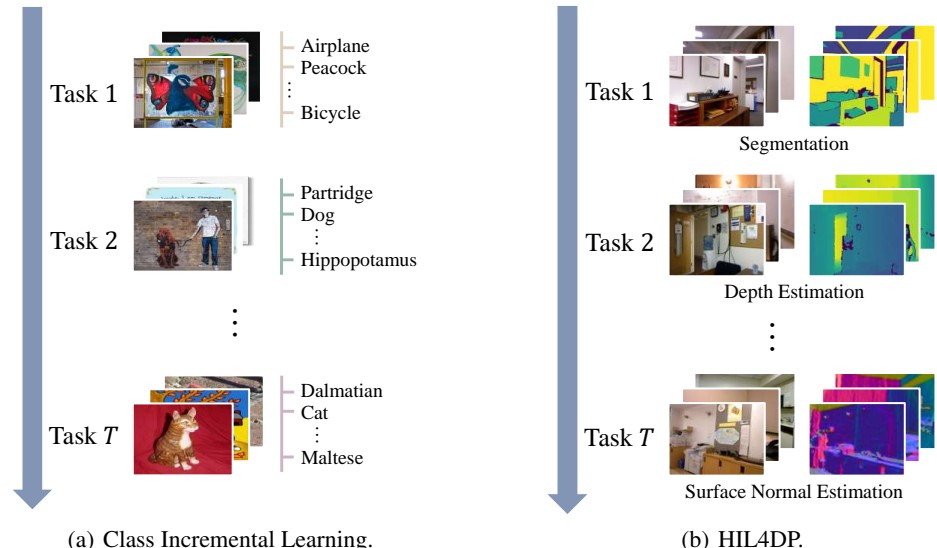

Figure 5: Comparison between Class Incremental Learning (CIL) and Heterogeneous Incremental Learning for Dense Prediction (HIL4DP). (a) CIL incrementally recognizes all encountered classes, while (b) HIL4DP progressively addresses all encountered heterogeneous dense prediction tasks.

In this section, we analyze the similarities and differences between the proposed heterogeneous incremental learning for dense prediction (HIL4DP) scenario and the traditional incremental learning (IL) scenario.

**Similarities.** The proposed HIL4DP scenario shares three key similarities with the traditional IL scenario: objectives, settings, and challenges. First, both HIL4DP and IL aim to achieve performance on sequential tasks comparable to that of joint training across multiple tasks (Wang et al., 2024). Second, in both settings, models can be trained for multiple epochs on all data for a given task, while data from previous and future tasks remains inaccessible. Finally, both HIL4DP and IL face the issue of catastrophic forgetting, where training on the current task leads to the loss of knowledge from previous tasks.

Table 5: Comparison between different categories within incremental learning.

| Subcategory | Domain Gap | Task ID | Multiple Task Type |
|---|---|---|---|
| CIL | × | × | × |
| TIL | × | √ | × |
| DIL | √ | × | × |
| HIL/HIL4DP | × | × | √ |

**Differences.** Traditional incremental learning can be classified into three subcategories: class incremental learning (CIL), task incremental learning (TIL), and domain incremental learning (DIL). Compared with other subcategories, the domain of the training data in DIL varies across tasks, while the number of classes remains consistent across different tasks. In both CIL and TIL, the domain of the training data remains consistent; however, as the number of tasks increases, so does the total number of classification categories. The key difference between TIL and CIL is that TIL requires a task ID during inference. However, CIL, TIL, and DIL remain restricted to classification tasks and do not support scenarios involving sequentially arriving heterogeneous tasks. In contrast, the proposed HIL4DP assumes tasks share an input distribution but differ in output types (*e.g.*, class labels and continuous values), which introduces unique challenges for HIL4DP. Illustration of the comparison can be found in Fig. 5 and Tab. 5.

## B  CHALLENGES OF HIL4DP

The unique challenges posed by the proposed HIL4DP scenario are listed as follows.

**Heterogeneous tasks.** The technical challenges inherent in HIL4DP are beyond those typically encountered in conventional IL scenarios. Different from traditional settings that focus on a single type of tasks (*e.g.*, classification or segmentation), HIL4DP requires learning different types of tasks at different training phases, where each task often involves distinct objective functions and heterogeneous outputs. This results in a more complex and challenging training process.

**Heterogeneous knowledge.** Different tasks require distinct and heterogeneous knowledge representations. For example, the depth estimation task requires a comprehensive understanding of 3D scenes, while the semantic segmentation task primarily relies on high-level structured semantic knowledge (Kim et al., 2023; Zhaoyun et al., 2022). This divergence presents a challenge for mitigating catastrophic forgetting during the learning of new tasks, highlighting the necessity of strategies that facilitate effective knowledge transfer across heterogeneous tasks.

**Fine-grained information.** DP tasks involve producing pixel-level outputs that rely on rich fine-grained information, thereby posing additional challenges (Zuo et al., 2022). This complexity makes retaining previously learned knowledge particularly difficult, requiring strategies capable of preserving fine-grained representations and producing globally coherent outputs across sequential DP tasks.

## C  EXPERIMENT DETAILS

### C.1  DETAILS OF DATASETS

To evaluate the performance of the proposed HISD, we conduct experiments on four datasets using different task numbers as different scenarios: *NYUv2* dataset for 3 tasks, *CityScapes* dataset for 2 tasks, *PASCAL-Context* dataset for 4 tasks, and *Taskonomy* dataset for 10 tasks.

*NYUv2* **dataset.** This dataset contains 795 training images and 654 testing images in a variety of indoor scenes with ground truth for three tasks (*i.e.*, 13-class semantic segmentation, depth estimation, and surface normal prediction). We use the mean Intersection over Union (mIoU) and Pixel Accuracy (Pix Arr) to evaluate the semantic segmentation task, and use the Absolute Error (Abs Err) and the Real Error (Rel Err) to evaluate the depth prediction task. For the surface normal estimation task, it is evaluated with the mean and the median of angular error measured in degrees, and the percentage of pixels whose angular error is within 11.25, 22.5, and 30 degrees.

*CityScapes* **dataset.** This dataset comprises 2,975 images for training and an additional 500 images for testing, where we conduct experiments on two tasks (*i.e.*, 7-class semantic segmentation and depth estimation). We use the mean Intersection over Union (mIoU) and Pixel Accuracy (Pix Arr) to evaluate the semantic segmentation task, and use the Absolute Error (Abs Err) and the Real Error (Rel Err) to evaluate the depth prediction task.

*PASCAL-Context* **dataset.** This dataset has 4,998 annotated training images and 5,105 annotated test images for four dense prediction tasks, including semantic segmentation, human parsing, surface normal estimation, and salience detection. The mIoU is used to evaluate the semantic segmentation task, human parts segmentation task, and saliency estimation task, while the mean of angular error measured in degrees is used to evaluate the surface normal estimation task.

*Taskonomy* **dataset.** We split the 1,390 images from three different views in this dataset into training data for the 10 tasks, reserving one unseen view for testing. The evaluation metric used for performance assessment is the test loss.

### C.2  BASELINES

We compare the proposed HISD method against vanilla training, as well as three categories of traditional IL methods: regularization-based methods including EWC (Kirkpatrick et al., 2017), LWF (Li & Hoiem, 2017), and SGP (Saha & Roy, 2023), which constrain the changes in important parameters, representations, and gradients; replay-based methods such as iCaRL (Rebuffi et al., 2017) and DER (Buzzega et al., 2020), which store historical data in a fixed-size memory and replay

them during the learning of new tasks; and the parameter isolation method SPG (Konishi et al., 2023), which combines orthogonal gradient projections with scaled gradient steps in the important gradient spaces for past tasks. For all replay-based methods, the exemplar size is fixed at 50.

For different methods, we perform grid searches on hyperparameters and select the best result. The hyperparameters of each method for different datasets are shown in Tab. 6.

Table 6: Hyperparameter of different methods.

| Method | *NYUv2* | | *CityScapes* | | *PASCAL-Context* | *Taskonomy* |
| | *Resnet-18* | *Resnet-50* | *Resnet-18* | *Resnet-50* | *Resnet-18* | *Resnet-18* |
|---|---|---|---|---|---|---|
| EWC | $10^9$ | $10^{10}$ | $10^6$ | $10^3$ | $10^6$ | $10^9$ |
| iCaRL | 0.01 | 0.1 | 0.01 | 0.1 | 0.1 | 1 |
| LWF | 5 | 5 | 0.01 | 0.1 | 5 | 0.1 |
| DER | 0.01 | 0.1 | 1 | 1 | 0.1 | 0.1 |
| SGP | 0.1 | 10 | 10 | 1000 | 1000 | 100 |

## C.3 IMPLEMENTATION DETAILS

The task sequences are randomly selected. For the *NYUv2* dataset, the sequence is: Semantic segmentation → Depth estimation → Surface normal prediction, as shown in Tab. 1. For the *Taskonomy* dataset, the sequence is: Semantic segmentation (Seg.) → Depth estimation (Dep.) → Surface normal estimation (Normal) → Reshading (Res.) → Keypoint-2D detection (K.-2D) → Edge-2D detection (E.-2D) → Euclidean distance (E.D.) → Curvatures (Curv.) → Keypoint-3D detection (K.-3D) → Edge-3D detection (E.-3D), as shown in Tab. 2.

For all methods, we adopt the following common settings to ensure a fair comparison. The batch size is set to 64 for the *CityScapes* dataset, 16 for the *Taskonomy* dataset, 48 for *NYUv2* and *PASCAL-Context* datasets. We use the Adam optimizer with an initial learning rate of $10^{-4}$, and adopt a linear learning rate scheduler with a warmup phase, where the warmup rate is set to 0.5. Weight decay is fixed at $10^{-5}$.

In the proposed HISD method, we perform grid searches for the hyperparameters $\alpha$, $k$, and $\tau$. Specifically, we set hyperparameters as follows: $\alpha = 3, k = 0.5, \tau = 0.9$ for *NYUv2* dataset on *Resnet-18*, $\alpha = 20, k = 0.6, \tau = 0.6$ for *NYUv2* dataset on *Resnet-50*, $\alpha = 100, k = 0.6, \tau = 0.6$ for *CityScapes* dataset on *Resnet-18*, $\alpha = 1, k = 0.8, \tau = 0.5$ for *CityScapes* dataset on *Resnet-50*, and $\alpha = 50, k = 0.5, \tau = 0.5$ for *PASCAL-Context* dataset, $\alpha = 1, k = 0.5, \tau = 0.9$ for *Taskonomy* dataset. We use the task-specific loss function as the per-pixel self-distillation loss function $\mathcal{L}_{\text{dis},j}$ of each task $\mathcal{T}_j$, *i.e.*, $\mathcal{L}_{\text{dis},j} = \mathcal{L}_j$. All methods are implemented using Pytorch framework, and all models are trained on RTX V100 GPUs.

## D ADDITIONAL RESULTS

Results for the shuffled task sequence on the *NYUv2* dataset are provided in Tab. 7. When the task sequence is: Surface normal prediction → Depth estimation → Semantic segmentation, the proposed HISD method consistently outperforms the baseline methods, further demonstrating that its effectiveness is independent of task sequence.

The results of the *CityScapes* dataset using *ResNet-18* with different task sequences are provided in Tab. 8. Tab. 9 presents the results of the proposed HISD method on the same dataset using *ResNet-50*, with semantic segmentation as the first task and depth estimation as the second. As can be seen, the proposed HISD method outperforms baseline methods in both mitigating the performance degradation of the previous task and improving overall performance. Note that although the *Cityscapes* dataset contains only 2 tasks, the HIL4DP scenario is fundamentally different from transfer learning (TL), as HIL4DP treats all tasks equally by preserving the performance of previous tasks. In contrast, TL primarily focuses on optimizing the performance of the target task.

Tab. 10 presents the results of the proposed method on the *PASCAL-Context* dataset, using *ResNet-18*. The tasks are trained sequentially in the sequence: Semantic Segmentation (Seg.) → Human

Table 7: Performance on the *NYUv2* dataset with a shuffled task sequence after the last training phase. The best results for each task are shown in **bold**. ↑(↓) means that the higher (lower) the value, the better the performance.

| Method | Segmentation | | Depth | | Surface Normal | | | | | $\Delta_b^T \uparrow$ |
|---|---|---|---|---|---|---|---|---|---|---|
| | | | | | Angle Distance | | Within $t°$ | | | |
| | mIoU↑ | Pix Acc↑ | Abs Err ↓ | Rel Err↓ | Mean ↓ | Median ↓ | 11.25 ↑ | 22.5 ↑ | 30 ↑ | |
| Vanilla training | 33.77 | 60.36 | 1.0261 | 0.3592 | 40.76 | 34.74 | 10.16 | 30.98 | 43.17 | +0.00% |
| Joint training | 41.84 | 66.14 | 0.5793 | 0.2201 | 31.53 | 25.78 | 22.38 | 44.54 | 56.36 | +3.09% |
| EWC | 29.78 | 56.75 | 0.9217 | 0.3218 | 39.15 | 33.31 | 10.48 | 31.98 | 44.95 | −0.77% |
| iCaRL | 23.87 | 53.78 | 1.5976 | 0.5474 | 35.87 | 33.19 | 11.60 | 31.55 | 44.63 | −27.11% |
| LwF | 31.49 | 58.96 | 0.8586 | 0.3044 | 37.66 | 32.54 | 11.76 | 33.50 | 46.17 | +0.77% |
| DER | 24.41 | 54.38 | 1.5884 | 0.5428 | 35.78 | 33.00 | 11.71 | 31.84 | 44.96 | −26.55% |
| SPG | **34.40** | 60.79 | 1.0025 | 0.3454 | 40.01 | 34.18 | 11.10 | 31.86 | 43.96 | +0.29% |
| SGP | 34.34 | **61.16** | 1.0859 | 0.3803 | 41.30 | 35.83 | 9.62 | 29.30 | 41.47 | −0.17% |
| HISD | 28.86 | 56.83 | **0.7024** | **0.2553** | **35.05** | **30.36** | **12.71** | **36.04** | **49.41** | **+1.18%** |

Table 8: Performance on two tasks after the last training phase (i.e., 7-class semantic segmentation and depth estimation) of the *CityScapes* dataset under two different sequences.

| Method | Segmentation → Depth | | | | | Depth → Segmentation | | | | |
|---|---|---|---|---|---|---|---|---|---|---|
| | Segmentation | | Depth | | $\Delta_b^T \uparrow$ | Segmentation | | Depth | | $\Delta_b^T \uparrow$ |
| | mIoU↑ | Pix Acc↑ | Abs Err ↓ | Rel Err↓ | | mIoU↑ | Pix Acc↑ | Abs Err ↓ | Rel Err↓ | |
| Vanilla training | 58.40 | 86.84 | 0.0203 | 50.0861 | +0.00% | 68.44 | 91.45 | 0.0456 | 77.8347 | +0.00% |
| Joint training | 71.38 | 92.15 | 0.0164 | 43.7236 | +15.06% | 71.38 | 92.15 | 0.0164 | 43.7236 | +28.23% |
| EWC | 66.14 | 90.01 | 0.0203 | 56.8526 | +0.85% | 65.56 | 90.06 | 0.0217 | 51.9560 | +19.98% |
| iCaRL | 67.10 | 91.09 | 0.0204 | 50.2167 | +4.76% | 57.24 | 87.82 | 0.0221 | 58.8481 | +13.90% |
| LwF | 62.67 | 87.98 | 0.0202 | **47.6400** | +3.50% | 67.52 | 89.54 | 0.0192 | 47.2005 | +24.91% |
| DER | 67.15 | 91.11 | 0.0206 | 51.3117 | +3.99% | 69.20 | **91.90** | 0.0256 | 54.3059 | +18.92% |
| SPG | 68.07 | **91.31** | 0.0484 | 94.8094 | −51.50% | **69.68** | 91.75 | 0.0418 | 104.0395 | −5.80% |
| SGP | 53.00 | 82.38 | 0.0202 | 49.2638 | −3.06% | 68.07 | 91.31 | 0.0484 | 94.8094 | −7.16% |
| HISD | **68.28** | 90.76 | **0.0192** | 51.7254 | **+5.89%** | 69.12 | 91.12 | **0.0186** | **45.5291** | **+26.70%** |

Table 9: Performance on two tasks after the last training phase (i.e., 7-class semantic segmentation and depth estimation) of the *CityScapes* dataset using *Resnet-50*. The best results for each task are shown in **bold**. ↑(↓) means that the higher (lower) the value, the better the performance.

| Method | Segmentation | | Depth | | $\Delta_b^T \uparrow$ |
|---|---|---|---|---|---|
| | mIoU↑ | Pix Acc↑ | Abs Err ↓ | Rel Err↓ | |
| Vanilla training | 64.93 | 88.93 | 0.0168 | 40.0604 | +0.00% |
| Joint training | 76.49 | 93.91 | 0.0155 | 45.7162 | +4.26% |
| EWC | 69.44 | 90.60 | **0.0154** | **41.4725** | +3.41% |
| iCaRL | 72.80 | 92.60 | 0.0166 | 47.3969 | −0.22% |
| LwF | 76.07 | 93.66 | 0.0175 | 43.5094 | +2.42% |
| DER | 73.26 | 93.16 | 0.0159 | 46.5341 | +1.70% |
| SPG | 57.99 | 85.69 | 0.0155 | 44.7024 | −4.55% |
| SGP | 65.53 | 88.44 | 0.0157 | 43.8087 | −0.61% |
| HISD | **76.52** | **93.81** | 0.0165 | 44.2749 | **+3.65%** |

Parsing (H.Parts) → Saliency Map (Sal.) → Surface Normal Estimation (Normal). As can be seen, the proposed HISD method outperforms baseline methods, demonstrating its superior performance.

# E  ADVANTAGES OF THE HIL4DP SCENARIO

We demonstrate the advantages of the HIL4DP scenario by comparing it with training separate task-specific models under the same amount of labeled data. Beyond its practical benefit of supporting heterogeneous tasks within a shared encoder, HIL4DP offers two key advantages: higher performance and lower memory overhead.

Table 10: Performance on four tasks (i.e., 21-class semantic segmentation, 7-class human parts segmentation, saliency estimation, and surface normal estimation) in the *PASCAL-Context* dataset. The best results for each task are shown in **bold**. ↑(↓) means that the higher (lower) the value, the better the performance.

| Method | Seg.↑ | H.Parts↑ | Sal.↑ | Normal↓ | $\Delta_b^T$ ↑ |
|---|---|---|---|---|---|
| Vanilla training | 11.88 | 27.40 | 52.09 | 21.4409 | $+0.00\%$ |
| Joint training | 57.77 | 48.05 | 60.88 | 21.0313 | $+60.05\%$ |
| EWC | 34.27 | **37.04** | 55.05 | 23.9992 | $+27.18\%$ |
| iCaRL | 12.06 | 30.17 | **56.84** | 20.7354 | $+3.00\%$ |
| LWF | 38.10 | 34.63 | 54.04 | 24.0761 | $+29.82\%$ |
| DER | 10.81 | 32.03 | 55.48 | **20.4669** | $+2.37\%$ |
| SPG | 7.16 | 20.31 | 47.38 | 20.5474 | $-8.81\%$ |
| SGP | 8.46 | 29.55 | 49.83 | 20.6048 | $-2.67\%$ |
| HISD | **39.96** | 33.62 | 53.76 | 23.7274 | $+\mathbf{31.45\%}$ |

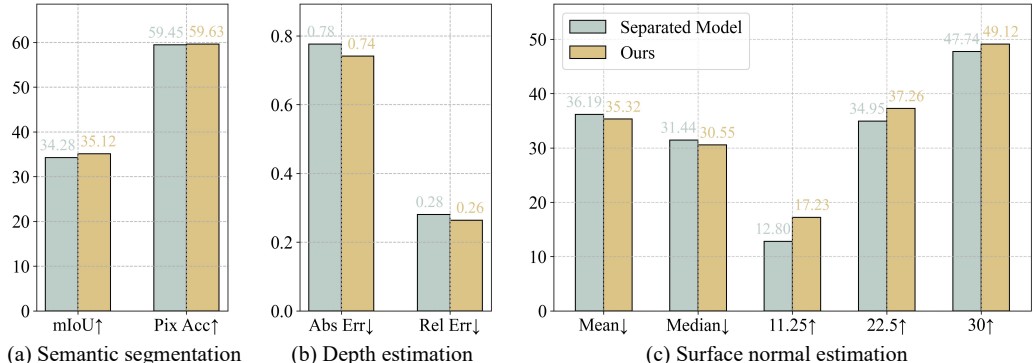

(a) Semantic segmentation    (b) Depth estimation    (c) Surface normal estimation

Figure 6: The comparison between training separate models and the proposed HISD. Each figure illustrates the performance improvement of the HISD method in the HIL4DP scenario of a given task. The symbol ↑ (↓) signifies that a higher (lower) value denotes better performance.

**Better performance.** The comparison on *NYUv2* dataset is shown in Fig. 6. As can be seen, HIL4DP achieves average improvements of 1.38%, 5.2%, and 9.87% on the semantic segmentation, depth estimation, and surface normal estimation tasks, respectively, compared to task-specific models.

**Lower memory overhead.** Compared to using a shared encoder in the HIL4DP scenario, training separate task-specific models introduces more parameter overhead of 172.23% for additional encoders when using *ResNet-18* with ten tasks. Notably, this overhead grows with both the encoder complexity and the number of tasks, which is efficiently avoided by the proposed HIL4DP scenario.

## F   USE OF LARGE LANGUAGE MODELS

Large language models (LLMs) were used solely for the purpose of improving the readability and language of this manuscript. This work was conceived, designed, and executed entirely by the authors without technical contribution from LLMs. The authors take full responsibility for all technical contributions presented in this manuscript.

