# OpenReview forum: "Heterogeneous Incremental Learning for Dense Prediction: Advancing Knowledge Retention via Self-Distillation"
_ICLR.cc/2026/Conference — ICLR 2026 Conference Withdrawn Submission_

### Official Review · Reviewer_ay4B · 2025-10-26

**Soundness:** 3
**Presentation:** 3
**Contribution:** 2
**Rating:** 4
**Confidence:** 4

**Summary:**

This paper introduces a new task-incremental learning (TIL) scenario for dense prediction tasks in computer vision, where each task shares the same input space X but has a distinct label space Y. To mitigate performance degradation in this continual learning setup, the authors propose HISD (Heterogeneity-aware Incremental Self-Distillation), which extends the classical knowledge distillation (KD)-based incremental learning framework originally popularized by LwF. Specifically, HISD retains knowledge from previous tasks by generating pseudo-labels using data from newly introduced tasks, thereby preserving old information without direct access to past data.
Building on this KD-based framework, the authors introduce two additional loss components, Distribution-Balanced Incremental Self-Distillation (DB-ISD) and Salience-Guided Incremental Self-Distillation (SG-ISD), which effectively enhance model performance, as supported by their ablation studies. Compared with representative incremental learning baselines such as EWC, LwF, and DER, the proposed HISD achieves the highest average performance gain over standard (non-incremental) training, demonstrating its efficacy in handling task heterogeneity in dense prediction settings.

**Strengths:**

1. The paper is generally well written, with clear explanations supported by descriptive examples and illustrative figures.
2. This seems to be the first study to address the proposed HIL problem within the context of dense prediction.
3. In Section 4.2, which presents the main technical contributions, the authors introduce two novel loss terms that extend the standard KD-based incremental learning framework. These components are shown to be effective according to the ablation studies.
4. The experiments are comprehensive, as the authors evaluate their method across varying numbers of continual tasks (i.e., 3, 5, and 10), which strengthens the validity of their claims.

**Weaknesses:**

1. The proposed method is specifically designed for and evaluated only on dense prediction tasks. It would be valuable to examine whether the proposed loss terms can generalize to other task types beyond this domain.
2. he performance improvement over existing baselines appears modest. Even compared with traditional methods such as EWC, the gains are not particularly substantial. In certain cases, especially regression-based tasks, HISD seems to struggle to achieve satisfactory results, although its overall averaged performance remains the best among compared methods.
3. Despite its empirical effectiveness, the paper lacks a clear theoretical or even intuitive justification for the design choice of using the geometric mean of group-wise losses. A deeper explanation or motivation for such design decisions would make the paper more convincing and insightful, allowing readers to better understand the reasoning behind the method rather than merely its implementation.

**Questions:**

1. My main concern is whether the reported performance gain of HISD stems partly from architectural differences between compared methods. Specifically, HISD appears to employ a branched architecture (one shared encoder with multiple task-specific decoders). Was this same architecture also applied to all baseline methods (e.g., iCaRL, EWC, etc.)? If not—if the baselines used their original single-branch architectures with multiple heads—an ablation study comparing different architectural settings would be necessary to ensure a fair comparison.
2. Related to the above, do all task-specific decoders in HISD have the same model capacity? If so (or if not), please clarify their sizes. Can these decoders be considered equivalent to the simple multi-head configurations commonly used in class-incremental learning (e.g., linear classifiers per task)?

---

### Official Review · Reviewer_SPdh · 2025-10-29

**Soundness:** 3
**Presentation:** 3
**Contribution:** 3
**Rating:** 4
**Confidence:** 3

**Summary:**

This paper tackles the new problem for continiual learning named as Heterogenous Incremental Learning for Dense Prediction. Existing methods are concentrating on single task but this paper focuses on the scenario, that can happen in real world, which encounters multiple different tasks. By analysing both heterogenuity and the property of dense prediction, the authors provide novel method that can handle this novel problem as well.

**Strengths:**

- This paper extends the scope of continual learning from homogeneous task to heterogeneous tasks which can be more realistic and also give intuition to researchers in this field.
- The authors analyse the proposed problem in multiple senses and propse multiple components that can effectively handle those aspects.
- Paper is easy to follow and well written. Related works seem to be well investigated.

**Weaknesses:**

- If we need to retain the multiple decoders for multiple tasks, what happens if we have different network for each task and apply existing incremental learning methods for each task/network?
- What happens if there are more than 2 same tasks in sequences? In that case, I am wondering about the performance degradation after encountering same task.
- One major concern across the entire continual learning literature is about memory/computational cost concern. Can you give detail about the memory/computational consumption of the proposed method compared to the other methods?
- Contribution of each components is only explained by Table 4. Can you elaborate more and give intuitive examples to better understand them?

Overall, I appreciate the authors' work but I have some concerns as stated above. I would be happy to adjust my score if the concerns are well resolved.

**Questions:**

See weaknesses

---

### Official Review · Reviewer_ZDfj · 2025-11-01

**Soundness:** 2
**Presentation:** 3
**Contribution:** 2
**Rating:** 2
**Confidence:** 4

**Summary:**

This paper introduces a new IL scenario termed Heterogeneous Incremental Learning (HIL) limited in Dense Prediction (HIL4DP), where tasks with different output types, such as segmentation, depth estimation, and surface normal prediction, arrive sequentially. To address this setting, the authors propose Heterogeneity-aware Incremental Self-Distillation (HISD), an exemplar-free framework that uses a shared encoder with task-specific decoders and introduces two additional loss functions: a distribution-balanced distillation loss and an edge-aware salience-guided loss. The method aims to retain knowledge across heterogeneous tasks through two self-distillation loss functions. The paper presents experiments on several dense prediction datasets showing moderate improvements over conventional IL baselines. While the proposed scenario is conceptually interesting, the overall novelty and empirical improvements remain modest.

**Strengths:**

1. The concept of Heterogeneous Incremental Learning (HIL) expands the scope of IL beyond conventional homogeneous settings, introducing a new perspective on learning across tasks with different output structures.

2. The use of a shared backbone with task-specific decoders is a reasonable and well-structured design choice. The proposed distribution-balanced and salience-guided self-distillation losses are clearly motivated, addressing pseudo-label imbalance and emphasizing edge-aware retention.

3. The paper is well-organized and clearly written, presenting the methodology, setup, and experimental results in a coherent manner. The experiments span multiple dense prediction datasets and demonstrate consistent improvements over existing baselines.

**Weaknesses:**

1. Since the proposed framework relies on task-specific decoders, it will require a task-id to gate the encoded features to the appropriate decoder at inference. This limits the proposed scenario similarly to a task-incremental setup, reducing its reality and scalability.

2. Both proposed losses, distribution-balanced salience-guided self-distillation, appear to overlap significantly with existing work in distillation [1-3], rather than fundamentally new mechanisms. Moreover, the overall process of the method is very similar to LwF: a shared encoder and a task-specific decoder with "L_new" loss and "distillation loss" using the previous network as the teacher model.

3. Each task employs a distinct loss, which contradicts the claim of a unified heterogeneous incremental learner (line 153) and reduces the realism of “heterogeneity-agnostic” learning.

4. Results are shown for only one or two task orders per dataset, undermining claims of sequence-agnostic generalization. As shown in Table 1 of the main paper and Table 7 of the Appendix, the average relative improvement exhibits a significant gap.

5. All experiments use CNN-based ResNet backbones, ignoring modern trends in IL that leverage pre-trained ViTs and parameter-efficient fine-tuning techniques such as Adapters, Prompts [4-6]. This omission limits the paper’s contemporary relevance.

6. The paper doesn’t convincingly explain why a model must sequentially learn heterogeneous tasks or why dense prediction specifically is chosen to instantiate HIL. The scenario lacks grounding in realistic applications (e.g., embodied agents or autonomous driving models).

7. The paper introduces several new hyperparameters (α, k, τ) but provides no sensitivity analysis, making the robustness of the method unclear. Although the authors said a grid search is performed for the hyperparameters, the range of their values is very wide (especially alpha, which controls the influence of loss for new knowledge and loss for distillation), as shown in Appendix C.3.

8. Omission of recent SOTA baselines reduces confidence. The chosen baselines (EWC, LwF, iCaRL, DER, SPG, SGP) are standard, but some are dated.

9. The main metric, average relative improvement Δ_b^T, measures only final performance against a vanilla baseline, overlooking the temporal dynamics central to continual learning. Since forgetting and forward transfer require intermediate evaluations to assess knowledge retention and reuse, Δ_b^T alone cannot reveal whether gains reflect true continual learning or merely final-task optimization.

10. The paper explicitly states that it “employs a task-shared encoder to acquire knowledge from a sequence of tasks and capture fine-grained features from images” (line 214) and that it serves as “a task-shared encoder that generalizes across tasks.” (line 216) However, despite positioning this shared encoder as the central mechanism of heterogeneous knowledge accumulation, the paper provides no empirical or theoretical analysis validating this role.

[1] Li, Ruihuang, et al. "Class-balanced pixel-level self-labeling for domain adaptive semantic segmentation." CVPR 2022.\
[2] Truong, Thanh-Dat, et al. "Fairness continual learning approach to semantic scene understanding in open-world environments." NeurIPS 2023.\
[3] Zhu, et al. "The edge of depth: Explicit constraints between segmentation and depth." CVPR 2020.\
[4] Wang, Zifeng, et al. "Learning to prompt for continual learning." CVPR 2022.\
[5] Park, et al. "Versatile Incremental Learning: Towards Class and Domain-Agnostic Incremental Learning." ECCV 2024.\
[6] Lin, et al. "Towards Continual Universal Segmentation." CVPR 2025.

**Questions:**

1. How does the proposed framework operate when the task identity is unknown at inference? Can the model infer or route inputs to the correct decoder, and if not, how does this align with the goal of scalable heterogeneous incremental learning?

2. Please clarify how the proposed distribution-balanced and salience-guided self-distillation losses differ fundamentally from prior works.

3. Could the authors report results averaged over multiple random task sequences and include standard deviations to demonstrate sequence-agnostic robustness? If performance varies substantially across orders, what factors contribute most to that sensitivity?

4. Have the authors evaluated the proposed method on a pre-trained ViT backbone using parameter-efficient tuning methods (e.g., Adapters, Prompts)? If not, how do the authors expect HISD to behave under such architectures, and what challenges might arise from transformer-based representations?

5. What are the real-world scenarios motivating sequential heterogeneous tasks? Why is dense prediction specifically chosen to instantiate this scenario rather than classification or multi-modal tasks?

6. Could the authors provide a detailed sensitivity analysis for α, k, τ, showing performance trends across different values? How stable is the method under moderate deviations from the reported grid-search configurations?

7. Could the authors provide results against at least one recent IL baseline?

8. Could the authors supplement the average relative improvement Δ_b^T metric with standard continual learning metrics such as Forgetting and Forward Transfer (FWT)? These would more directly capture temporal knowledge dynamics rather than only final-stage performance.

9. The paper claims that the shared encoder “acquires knowledge from a sequence of tasks” and “generalizes across tasks,” yet provides no empirical evidence supporting this. Could the authors include representational analyses (e.g., t-SNE, subspace overlap, or CKA similarity) to demonstrate whether the encoder indeed accumulates task-agnostic representations across training?

---

### Official Review · Reviewer_QbAg · 2025-11-01

**Soundness:** 3
**Presentation:** 3
**Contribution:** 3
**Rating:** 4
**Confidence:** 3

**Summary:**

This paper addresses the problem of heterogeneous incremental learning for dense prediction tasks, where the learner must preserve heterogeneous knowledge corresponding to different output space structures during incremental updates. To tackle this challenge, the authors propose the Heterogeneity-aware Incremental Self-Distillation (HISD) method, an exemplar-free framework that preserves previously learnt heterogeneous knowledge by self-distillation. Specifically, a distribution-balanced loss is introduced to alleviate the global imbalance in prediction distributions, while a salience-guided loss focuses learning on informative edge pixels extracted using the Sobel operator. Extensive experiments and ablation studies are conducted to validate the effectiveness of the proposed method and the contribution of each design.

**Strengths:**

+ The paper introduces a novel and practical scenario for incremental learning, heterogeneous incremental learning, where the incoming tasks are heterogeneous. The focus on dense prediction tasks makes the study relevant and potentially useful for real-world applications.
+ The paper is technically sound, and the proposed method is well-motivated to address the challenges of retaining heterogeneous knowledge during incremental learning.
+ Extensive experiments and ablation studies are conducted to validate the contributions and efficacy of each designed component in the proposed framework.

**Weaknesses:**

- A major concern is the scalability and computational cost of the proposed framework. Since HISD adds a new decoder for each task and performs self-distillation across all previous decoders, both the number of parameters and the training cost may grow rapidly with the number of tasks, limiting its practicality for large-scale or long-term incremental learning.
- One of the stated motivations is the difficulty of data collection for dense prediction tasks, yet the experiments do not fully demonstrate the benefits of heterogeneous incremental learning under realistic conditions. The experiments are conducted on the same dataset with different numbers of tasks, which are from the same domains and well-annotated. However, in practice, tasks often originate from different domains with varying data distributions and labeling quality. The absence of such cross-domain evaluations raises concerns about the robustness and generalizability of the proposed approach.
- The task-specific performance results are not entirely convincing. While the overall average performance outperforms SOTA methods, several individual tasks fail to achieve comparable results. This suggests that the overall gain may be dominated by a few tasks with significant improvement, while others perform suboptimally. This uneven performance pattern raises questions about the consistency and general effectiveness of the proposed method across diverse tasks.

**Questions:**

Please see the concerns in the weaknesses section.

---

### Note · Authors · 2025-11-14

I have read and agree with the venue's withdrawal policy on behalf of myself and my co-authors.